# Hierarchical Gaussian Process Priors for Bayesian Neural Network Weights

**Theofanis Karaletsos**[*]
Facebook
theokara@fb.com

**Thang D. Bui**[*]
Uber AI and University of Sydney
thang.bui@uber.com

## Abstract

Probabilistic neural networks are typically modeled with independent weight priors, which do not capture weight correlations in the prior and do not provide a parsimonious interface to express properties in function space. A desirable class of priors would represent weights compactly, capture correlations between weights, facilitate calibrated reasoning about uncertainty, and allow inclusion of prior knowledge about the function space such as periodicity or dependence on contexts such as inputs. To this end, this paper introduces two innovations: (i) a Gaussian process-based hierarchical model for network weights based on unit priors that can flexibly encode correlated weight structures, and (ii) input-dependent versions of these weight priors that can provide convenient ways to regularize the function space through the use of kernels defined on contextual inputs. We show these models provide desirable test-time uncertainty estimates on out-of-distribution data and demonstrate cases of modeling inductive biases for neural networks with kernels which help both interpolation and extrapolation from training data.

## 1   Introduction

Bayesian neural networks (BNNs) [see e.g. MacKay, 1992, Neal, 1993, Ghahramani, 2016] are one of the research frontiers on combining Bayesian inference and deep learning, potentially offering flexible modelling power with calibrated predictive performance. In essence, applying probabilistic inference to neural networks allows all plausible network parameters, not just the most likely, to be used for predictions. Despite the strong interest in the community for the exploration of BNNs, there remain unanswered questions: (i) how can we model neural network functions to encourage behaviors such as interpolation between signals and extrapolation from data in meaningful ways, for instance by encoding prior knowledge, or how to specify priors which facilitate uncertainty quantification, and (ii) many scalable approximate inference methods are not rich enough to capture complicated posterior correlations in large networks, resulting in undesirable predictive performance at test time.

This paper attempts to tackle some of the aforementioned limitations by taking inspiration from the Gaussian Process (GP) literature. First, we propose a hierarchical GP prior over weights to achieve a compact non-parametric representation of neural networks weights by utilizing unit-level latent variables. Second, we propose to utilize this non-parametric GP prior on weights to overcome a key limitation of current weight priors: their global nature. We explore the use of product kernels to implement input-dependence as a variation of the proposed prior, yielding models that have local (per-datapoint or per-layer) priors. We explore what effects these per-datapoint priors have in comparison to global priors on the network's ability to generalize. A structured variational inference approach is employed that side-steps the need to do inference in the weight space and amortizes per-datapoint weight inference. A consequence of our model setup and usage of approximate inference is that our model is parametrized compactly by inducing points akin to GPs, but in our case those are

---

[*]Both authors contributed equally. Work done while TK was at Uber AI

*inducing weights*. The proposed priors and approximate inference scheme are demonstrated to exhibit beneficial properties for tasks such as generalization and uncertainty quantification.

The paper is organized as follows: in Section 2 we review graph-based hierarchical modeling for BNNs. In Section 3 we introduce the global and local weight models and their applications to neural networks. Efficient inference algorithms are presented in Section 4, followed by a suite of experiment to validate their performance in Section 5. We review related work in Section 6.

## 2 Background: Unit-Priors For Neural Networks

Our work builds on graph-based hierarchical neural network models using latent variables associated with units in a network as proposed in [Karaletsos et al., 2018]. In that model, each unit (visible or hidden) of the $l$-th layer of the network has a corresponding latent hierarchical variable $\mathbf{z}_{l,i}$, of dimensions $D_z$, where $i$ denotes the index of the unit in a layer. Note that these latent variables do not describe the activation of units, but rather constitute latent features associated with a unit. The design of these latent variables is judiciously chosen to construct the weights in the network as follows: a weight in the $l$-th layer, $w_{l,i,j}$ is regressed by using the concatenation of latent variable $\mathbf{z}$'s of the $i$-th input unit and the $j$-th output unit as inputs of a mapping function $f\left(\left[\mathbf{z}_{l,i}, \mathbf{z}_{l+1,j}\right]\right)$.

We can summarize this relationship by introducing a set of *weight encodings* $\mathbf{C}_w(\mathbf{z})$, one for each individual weight in the network $\mathbf{c}_{w_{l,i,j}} = \left[\mathbf{z}_{l,i}, \mathbf{z}_{l+1,j}\right]$, which can be deterministically constructed from the collection of unit latent variable samples $\mathbf{z}$ by concatenating them correctly according to network architecture. The probabilistic description of the relationship between the weight codes (summarizing the structured latent variables) and the weights $\mathbf{w}$ is:

$$p(\mathbf{w}|\mathbf{z}) = p(\mathbf{w}|\mathbf{C}_w(\mathbf{z})) = \prod_{l=1}^{L-1} \prod_{i=1}^{H_l} \prod_{j=1}^{H_{l+1}} p(w_{l,i,j}|\mathbf{c}_{w_{l,i,j}}),$$

where $l$ denotes a visible or hidden layer and $H_l$ is the number of units in that layer, $L$ is the total number of layers in the network, and $\mathbf{w}$ denotes all the weights in this network.

In [Karaletsos et al., 2018], a small parametric neural network regression model (conceptually a *structured hyper-network*) is chosen to map the latent variables to the weights, using either a Gaussian noise model $p(\mathbf{w}|\mathbf{C}_w(\mathbf{z}), \theta) = \mathcal{N}(\mathbf{w}|\mathrm{NN}_\theta(\mathbf{C}_w(\mathbf{z})))$ or an implicit noise model: $p(\mathbf{w}|\mathbf{C}_w(\mathbf{z}), \theta) \propto \mathrm{NN}_\theta(\mathbf{C}_w(\mathbf{z}), \epsilon)$, where $\epsilon$ is a random variate. We will call this network a *meta mapping*. Note that given the collection of sampled unit variables $\mathbf{z}$ and the derived codes $\mathbf{C}_w(\mathbf{z})$, the weights (or theirs mean and variance) can be obtained efficiently in parallel. A prior over latent variables $\mathbf{z}$ completes the model specification, $p(\mathbf{z}) = \prod_{l=0}^{L} \prod_{i=0}^{H_l} \mathcal{N}(\mathbf{z}_{l,i}; \mathbf{0}, \mathbf{I})$. The joint density of the resulting hierarchical BNN is then specified as follows,

$$p(\mathbf{y}, \mathbf{w}, \mathbf{z}|\mathbf{x}, \theta) = p(\mathbf{z})p(\mathbf{w}|\mathbf{C}_w(\mathbf{z}), \theta) \prod_{n=1}^{N} p(\mathbf{y}_n|\mathbf{w}, \mathbf{x}_n),$$

with $N$ denoting the number of observation tuples $\{\mathbf{x}_n, \mathbf{y}_n\}$.

Variational inference was employed in prior work to infer $\mathbf{z}$ (and $\mathbf{w}$ implicitly), and to obtain a point estimate of $\theta$, as a by-product of optimising the variational lower bound. Critically, in this representation weights are only implicitly parametrized through the use of these latent variables, which transforms inference on weights into inference of the much smaller collection of latent unit variables.

The central motivations for our adoption of this parameterization are two-fold. *First*, the number of visible and hidden units in a neural network is typically much smaller than the number of weights. For example, for the $l$-th weight layer, there are $H_l \times H_{l+1}$ weights compared to $H_l + H_{l+1}$ associated latent variables. This encourages the development of models and inference schemes that can work in the lower-dimensional hierarchical latent space directly without the need to model individual weights privately. This structured representation per weight allows powerful hierarchical models to be used without requiring high dimensional parametrizations (i.e. hypernetworks for the entire weight tensor). Specifically, a GP-LVM prior [Lawrence, 2004] over all network weights appears infeasible without such an encoding of structure. *Second*, the compact latent space representation facilitates attempts at building fine-grained control into weight priors, such as structured prior knowledge as we will see in the following sections.

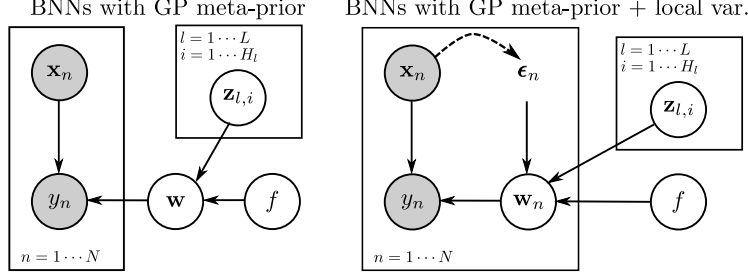

Figure 1: Graphical depiction of BNNs with hierarchical GP-MetaPriors and ones with input-dependent variables.

## 3 Hierarchical GP-Priors For BNN Weights

Notice that in Section 2, the mapping from the hierarchical latent variables to the weights is a parametric non-linear function, specified by a neural network. We replace the parametric neural network by a probabilistic functional mapping and place a nonparametric Gaussian process prior over this function. That is,

$$p(w_{l,i,j}|f, \mathbf{c}_{w_{l,i,j}}) = \mathcal{N}(w_{l,i,j}; f([\mathbf{z}_{l,i}, \mathbf{z}_{l+1,j}]), \sigma_w^2),$$
$$p(f|\gamma) = \mathcal{GP}(f; \mathbf{0}, k_w(\cdot, \cdot|\gamma)),$$

where we have assumed a zero-mean GP, $k_\gamma(\cdot, \cdot)$ is a covariance function and $\gamma$ is a small set of hyper-parameters, and a homoscedastic[2] Gaussian noise model with variance $\sigma_w^2$.

The effect is that the latent function introduces correlations for the individual weight predictions,

$$P(\mathbf{w}|\mathbf{z}) = P(\mathbf{w}|\mathbf{C}_w(\mathbf{z})) = \int_f p(f) \Big[ \prod_{l=1}^{L-1} \prod_{i=1}^{H_1} \prod_{j=1}^{H_{l+1}} p(w_{l,i,j}|f, \mathbf{z}_{l,i}, \mathbf{z}_{l+1,j}) \Big] df.$$

Notably, while the number of latent variables and weights can be large, the input dimension to the GP mapping is only $2D_z$, where $D_z$ is the dimensionality of each latent variable $\mathbf{z}$. The GP mapping effectively performs one-dimensional regression from latent variables to individual weights while capturing their correlations. We will refer to this mapping as a **GP-MetaPrior** (*MetaGP*). We define the following kernel at the example of two weights in the network,

$$k_w(c_{w_1}, c_{w_2}) = k_w([\mathbf{z}_{l^1,i^1}, \mathbf{z}_{l^1+1,j^1}], [\mathbf{z}_{l^2,i^2}, \mathbf{z}_{l^2+1,j^2}])$$

In this section and what follows, we will use the popular exponentiated quadratic (RBF) kernel with ARD lengthscales, $k(\mathbf{x}_1, \mathbf{x}_2) = \sigma_k^2 \exp\left(\sum_{d=1}^{2D_z} \frac{-(x_{1,d}-x_{2,d})^2}{2l_d^2}\right)$, where $\{l_d\}_{d=1}^{2D_z}$ are the lengthscales and $\sigma_k^2$ is the kernel variance.

**BNNs with GP-MetaPriors** are then specified by the following joint density over all variables:

$$p(\mathbf{y}, \mathbf{w}, \mathbf{z}, f|\mathbf{x}) = p(\mathbf{z})p(f)p(\mathbf{w}|f, \mathbf{z})p(\mathbf{y}|\mathbf{w}, \mathbf{x})$$
$$= p(\mathbf{z})p(f)p(\mathbf{w}|f, \mathbf{C}_w(\mathbf{z})) \prod_{n=1}^{N} [p(\mathbf{y}_n|\mathbf{w}, \mathbf{x}_n)].$$

An important task is marginalization of the latent quantities given data to perform posterior inference, which we discuss in Section 4.

### 3.1 Modeling Input-Dependent Weights With Auxiliary Kernels

While the hierarchical latent variables and meta mappings introduce non-trivial coupling between the weights a priori, they are inherently global. That is, a function drawn from the model, represented by

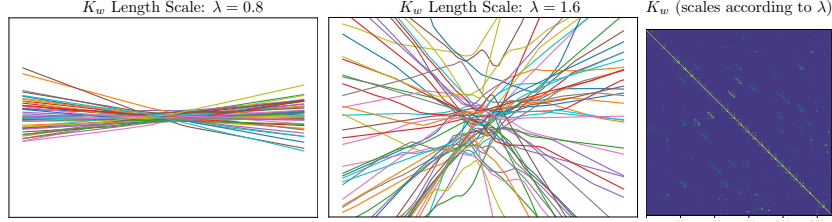

$K_w$ Length Scale: $\lambda = 0.8$     $K_w$ Length Scale: $\lambda = 1.6$     $K_w$ (scales according to $\lambda$)

Figure 2: **Illustration of MetaGP Prior Samples**: We show function samples generated from a [1,20,10,1] BNN with ReLUs. We draw one sample per unit $\mathbf{z}_{\text{viz}} \sim p(\mathbf{z})$, subsequently draw 40 function samples (individually colored) from the BNN by drawing from the conditional prior $\mathbf{w} \sim \mathcal{N}(\mathbf{w}|\mathbf{z}_{\text{viz}})$ and regressing $\mathbf{y} \sim p(\mathbf{y}|\mathbf{w}, \mathbf{x})$. (**left**) Samples drawn when the global RBF kernel for the GP has a length scale set to a small value. (**middle**) given samples $\mathbf{z}_{\text{viz}}$, we only change the length scale to be larger and visualize the functions induced by the BNN (**right**) the weight kernel given latent variables $\mathbf{z}_{\text{viz}}$. Other samples of $\mathbf{z}$ would induce different weight models $\mathcal{N}(\mathbf{w}|\mathbf{z}_{\text{viz}})$.

a set of weights, does not take into account the inputs at which the function will be evaluated. In this section, we will describe modifications to our weight prior which allow conditional weight models on inputs.

To this end, we introduce the input variable into the weight codes $\mathbf{c}_{w_{n,l,i,j}} = \left[\mathbf{c}_{w_{l,i,j}}, \mathbf{x}_n\right] = \left[\mathbf{z}_{l,i}, \mathbf{z}_{l+1,j}, \mathbf{x}_n\right]$, which we utilize to yield input-conditional weight models $p(w_{n,l,i,j}|f, \mathbf{z}_{l,i}, \mathbf{z}_{l+1,j}, \mathbf{x}_n)$ through the use of product kernels. Concretely, we introduce a new **input kernel** $k_{\text{aux}}$ which multiplied with the global weight kernel $k_w$ gives the kernel $k_{\text{local}}$ for the meta mapping,

$$k_{\text{local}}(\mathbf{c}_{w_1, x_1}, \mathbf{c}_{w_2, x_2}) = k_w(\mathbf{c}_{w_1}, \mathbf{c}_{w_2}) \cdot k_{\text{aux}}(\mathbf{x}_1, \mathbf{x}_2)$$

where $k_w$ is the kernel defined over latent-variable weight codes from Section 3 and $k_{\text{aux}}$ is an auxiliary kernel modeling input-dependence on $\mathbf{x}_n$. This factorization over kernels represents an assumption of separable influence on functions by latent variables $\mathbf{z}$ and inputs. The weight priors are now also local to each data point, in a similar vein to how functions are drawn from a GP, while still instantiating an explicit, weight-based model.

To scale this to large inputs, we learn transformations of inputs for the conditional weight model $\boldsymbol{\epsilon}_n = g(\mathbf{V}\mathbf{x}_n)$, for a learned mapping $\mathbf{V}$ and a nonlinearity $g$ and generalize weight codes to $\mathbf{c}_{w_{n,l,i,j}} = \left[\mathbf{z}_{l,i}, \mathbf{z}_{l+1,j}, \boldsymbol{\epsilon}_n\right]$, with $\mathbf{C}_{w,x}(\mathbf{z}, \mathbf{x})$ describing their collection. In detail, each auxiliary input is obtained via a (potentially nonlinear) transformation applied to an input: $\boldsymbol{\epsilon}_n = g(\mathbf{V}\mathbf{x}_n)$, where $\mathbf{V} \in \mathcal{R}^{D_{\text{aux}} \times D_x}$, and $D_{\text{aux}}$ and $D_x$ are the dimensionality of $\boldsymbol{\epsilon}_n$ and $\mathbf{x}_n$, respectively, and $g(\cdot)$ is an arbitrary transformation. We may also layer these transformations in general. We typically set $D_{\text{aux}} \ll D_x$ so this transformation could be thought of as a dimensionality reduction operation. For low dimensional inputs, we set $\boldsymbol{\epsilon}_n = \mathbf{x}_n$.

Including these transformations yields the weight model $p(w_{n,l,i,j}|f, \mathbf{z}, \mathbf{V}, \mathbf{x}_n) = \mathcal{N}(w_{n,l,i,j}; f(\mathbf{c}_{w_{n,l,i,j}}), \sigma_w^2)$, that is, the input dimension of the meta mapping is now $2D_z + D_{\text{aux}}$. Additionally, we also place a prior over the linear transformation: $p(\mathbf{V}) = \mathcal{N}(\mathbf{V}; \mathbf{0}, \mathbf{I})$. We will refer to this mapping as a **Local GP-MetaPrior** (*MetaGP-local*).

*What effects should we expect from such a modulation?* Consider the use of an exponentiated quadratic kernel: we would expect data which lies far away from training data to receive small kernel values from $K_{\text{aux}}$. This, in turn, would modulate the entire kernel $K_{\text{local}}$ for that data point to small values, leading to a weight model that reverts increasingly to the prior. We would expect such a model to help with modeling uncertainty by resetting weights to uninformative distributions away from training data. One may also want to use this mechanism to express inductive biases about the function space, such as adding structure to the weight prior that can be captured with a kernel. This is an appealing avenue, as multiple useful kernels have been found in the GP literature that allow modelers to describe relationships between data, but have previously not been accessible to neural network modelers. We consider this a novel form of functional regularization through the weight prior, which can imbue the entire network with structure that will constrain its function space.

**BNNs with Local GP-MetaPriors** specify neural networks with individual weight priors per data-point (also see Graphical Model in Fig. 1):

$$p(\cdot) = p(\mathbf{z})p(f)\prod_{n=1}^{N} p(\mathbf{y}_n, \mathbf{w}_n | f, \mathbf{z}, \mathbf{x}_n) = p(\mathbf{z})p(f)\prod_{n=1}^{N}\left[ p(\mathbf{w}_n | f, \mathbf{C}_w(\mathbf{z}, \mathbf{x}_n))p(\mathbf{y}_n | \mathbf{w}_n, \mathbf{x}_n)\right].$$

Inference and learning are modified accordingly as explained in Appendix C.

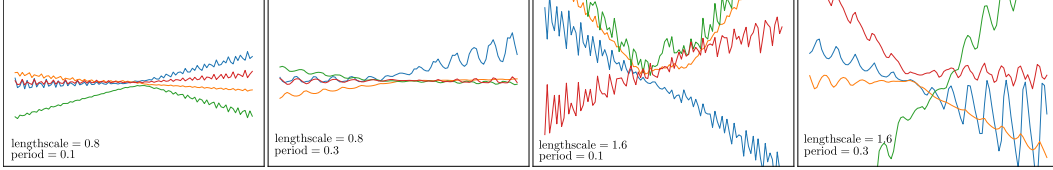

Figure 3: **Illustration MetaGP-Local Prior Samples**: We remind the reader that the input-dependent weight prior has a factorized kernel structure $k_{\text{local}} = k_w \cdot k_{\text{aux}}$, and demonstrate the effect of each kernel separately in terms of its effects on the weights for the neural network ([1,20,10,1]). Given the same samples $\mathbf{z}_{\text{viz}}$ and kernel parameter choices for $k_w$ as in Fig. 2, we only vary the period parameter for an auxiliary periodic kernel. **Left**: We show function samples using a $k_{\text{aux}}$ with period of 0.1 and 0.3 in combination with $k_w$ with length-scale 0.8. While the functions are still relatively flat, the auxiliary kernel induces weight priors which lead to periodic function samples consistent with the auxiliary kernel setting. **Right**: Similarly, for the $k_w$ with the larger lengthscale, we again observe periodic functions consistent with the set period, but see that the functions sampled have more variance, consistent with the larger length-scale of the weight-kernel $k_w$.

## 4 Inference and learning using stochastic structured variational inference

Performing inference is challenging due to the non-linearity of the neural network and the need to infer an entire latent function $f$. Here, we address these problems for our global weight prior *MetaGP*, deriving a structured variational inference scheme that makes use of innovations from inducing point GP approximation literature, utilizing learned inputs and function values $\{\mathbf{C_u}, \mathbf{u}\}$ which act as representative data points to parametrize the GP [Titsias, 2009, Hensman et al., 2013, Quiñonero-Candela and Rasmussen, 2005, Matthews et al., 2016, Bui et al., 2017], and previous work on inferring Unit-Priors [Karaletsos et al., 2018].

We first partition the space $\mathcal{C}$ of inputs (or *weight codes*) to the function $f$ into a finite set of $M$ variables called inducing inputs $\mathbf{C_u} = \{\mathbf{c}_{u,m}\}_{m=1}^{M}$ where $\mathbf{c}_{u,m} \in \mathcal{R}^{2D_z}$ and the remaining inputs, $\mathcal{C} = \{\mathbf{C_u}, \mathcal{C}_{\neq \mathbf{C_u}}\}$. The function $f$ is partitioned identically, $f = \{\mathbf{u}, f_{\neq \mathbf{u}}\}$, where $\mathbf{u} = f(\mathbf{C_u})$ are the *inducing weights*. We can then rewrite the GP prior as follows, $p(f) = p(f_{\neq \mathbf{u}} | \mathbf{u})p(\mathbf{u})$.[3] In particular, a variational approximation is judiciously chosen to mirror the form of the joint density: $q(\mathbf{w}, \mathbf{z}, f) = q(\mathbf{z})p(f_{\neq \mathbf{u}} | \mathbf{u})q(\mathbf{u})p(\mathbf{w} | f, \mathbf{z})$, where the variational distribution over $\mathbf{w}$ is made to explicitly depend on remaining variables through the conditional prior, and $q(\mathbf{z})$ is chosen to be a diagonal (mean-field) Gaussian density, $q(\mathbf{z}) = \mathcal{N}(\mathbf{z}; \boldsymbol{\mu}_\mathbf{z}, \text{diag}(\boldsymbol{\sigma}_\mathbf{z}^2))$, and $q(\mathbf{u})$ is chosen to be a correlated multivariate Gaussian, $q(\mathbf{u}) = \mathcal{N}(\mathbf{u}; \boldsymbol{\mu_u}, \Sigma_\mathbf{u})$. This approximation allows convenient cancellations yielding a tractable variational lower bound as follows,

$$\mathcal{F}(\cdot) = \int_{q(\mathbf{w},\mathbf{z},f)} \log \frac{p(\mathbf{z})\cancel{p(f_{\neq \mathbf{u}} | \mathbf{u})}p(\mathbf{u})\cancel{p(\mathbf{w}|f,\mathbf{z})}p(\mathbf{y}|\mathbf{w},\mathbf{x})}{q(\mathbf{z})\cancel{p(f_{\neq \mathbf{u}} | \mathbf{u})}q(\mathbf{u})\cancel{p(\mathbf{w}|f,\mathbf{z})}}$$

$$\approx -\text{KL}[q(\mathbf{z})||p(\mathbf{z})] - \text{KL}[q(\mathbf{u})||p(\mathbf{u})] + \frac{1}{S}\sum_{s=1}^{S}\int_{\mathbf{w},f} q(\mathbf{w}, f | \mathbf{z}_s)\log p(\mathbf{y}|\mathbf{w},\mathbf{x}),$$

where the last expectation has been approximated by simple Monte Carlo with the reparameterization trick, i.e. $\mathbf{z}_s \sim q(\mathbf{z})$ [Salimans and Knowles, 2013, Kingma and Welling, 2013, Titsias and Lázaro-Gredilla, 2014]. We will next discuss how to approximate the expectation

$\mathcal{F}_s = \int_{\mathbf{w}, f} q(\mathbf{w}, f | \mathbf{z}_s) \log p(\mathbf{y} | \mathbf{w}, \mathbf{x})$. Note that we split f into $f_{\neq \mathbf{u}}$ and $\mathbf{u}$, and that we can integrate $f_{\neq \mathbf{u}}$ out exactly to give, $q(\mathbf{w} | \mathbf{z}_s, \mathbf{u}) = \mathcal{N}(\mathbf{w}; \mathbf{A}^{(s)} \mathbf{u}, \mathbf{B}^{(s)})$,

$$\mathbf{A}^{(s)} = \mathbf{K}_{\mathbf{wu}}^{(s)} \mathbf{K}_{\mathbf{uu}}^{-1}, \qquad \mathbf{B}^{(s)} = \mathbf{K}_{\mathbf{ww}}^{(s)} - \mathbf{K}_{\mathbf{wu}}^{(s)} \mathbf{K}_{\mathbf{uu}}^{-1} \mathbf{K}_{\mathbf{uw}}^{(s)} + \sigma_w^2 \mathbf{I},$$

where $\mathbf{K}_{\mathbf{wu}}^{(s)} = k_w(\mathbf{C}_{\mathbf{w}}^{(s)}, \mathbf{C}_{\mathbf{u}})$, $\mathbf{K}_{\mathbf{uu}} = k_w(\mathbf{C}_{\mathbf{u}}, \mathbf{C}_{\mathbf{u}})$, $\mathbf{K}_{\mathbf{ww}}^{(s)} = k_w(\mathbf{C}_{\mathbf{w}}^{(s)}, \mathbf{C}_{\mathbf{w}}^{(s)})$. At this point, we can either (i) sample $\mathbf{u}$ from $q(\mathbf{u})$, or (ii) integrate $\mathbf{u}$ out analytically. Opting for the second approach gives $q(\mathbf{w} | \mathbf{z}_s) = \mathcal{N}(\mathbf{w}; \mathbf{A}^{(s)} \boldsymbol{\mu}_{\mathbf{u}}, \mathbf{B}^{(s)} + \mathbf{A}^{(s)} \Sigma_{\mathbf{u}} \mathbf{A}^{\top, (s)})$, the former just omits the second covariance term and uses a sample $\mathbf{u}^s$ for the predictive mean instead of $\boldsymbol{\mu}_{\mathbf{u}}$.

In contrast to GP regression and classification in which the likelihood term is factorized point-wise w.r.t. the parameters and thus their expectations only involve a low dimensional integral, we have to integrate out $\mathbf{w}$ which for GPs entails inversion of the $|\mathbf{w}| \times |\mathbf{w}|$ matrix $\mathbf{K}$ (which is $\mathbf{B}^{(s)}$ when we don't sample $\mathbf{u}$ or the full term above). This is feasible for small neural networks with up to a few thousand weights, but becomes intractable for more general architectures. In order to scale to larger networks, we introduce a diagonal approximation, which given a sample $\mathbf{u}^s$ looks as follows, $\hat{q}(\mathbf{w} | \mathbf{z}_s) = \mathcal{N}(\mathbf{w}; \mathbf{A}^{(s)} \mathbf{u}^s, \text{diag}(\mathbf{B}^{(s)}))$. Whilst the diagonal approximation above might look poor at first glance, it is conditioned on a sample of the latent variables $\mathbf{z}_s$ and thus the weights' correlations induced by the hierarchical unit-structure are retained after integrating out $\mathbf{z}$. Such correlations are illustrated in Fig. 11, showing the marginal and conditional covariance structures for the weights of a small neural network, separated into diagonal and full covariance models. We also provide a qualitative and quantitative analysis of performance of different approximations to $q(\mathbf{w} | \mathbf{z}_s)$ in the appendix, including the diagonal approximation presented here, and show that not only is this approximation fast but also that it performs competitively with full covariance models. The expected log-likelihood $\mathcal{F}_s$ is estimated by $\mathcal{F}_s \approx \frac{1}{J} \sum_{j=1}^{J} \log p(\mathbf{y} | \mathbf{w}_j, \mathbf{x})$ with samples $\mathbf{w}_j \sim \hat{q}(\mathbf{w} | \mathbf{z}_s)$ [4]. The final lower bound is then optimized to obtain the variational parameterers of $q(\mathbf{u})$, $q(\mathbf{z})$, and estimates for the noise in the meta-GP model, the kernel hyper-parameters and the inducing inputs.

In Appendix C we will highlight the modifications necessary to make this inference strategy work for *MetaGP-local* where we have a prior for each datapoint, exploiting that the inference strategy naturally permits amortization per datapoint and thus foregoing the need for test-time inference.

## 5 Experiments

In this section, we evaluate the proposed priors and inference scheme on several regression and classification datasets, and study effects of per-datapoint priors as we proposed on extrapolation, interpolation, and out-of-distribution data.[5] We use $M = 50$ inducing weights for all experiments in this section. Details for the experimental settings and additional experiments are included in the appendices.

We also explore the interplay of parametrization of kernels (per layer or per network kernels, amount of inducing weights used for global and local model) and properties of the dataset and the architecture in Sec. B.1, showing that the proposed models can be used flexibly in various scenarios.

Finally, in Sec. D Fig. 8 we provide an experiment clarifying the difference between a GP and the local variant of our model, which demonstrates that our model even with a mis-specified $k_{\text{aux}}$ can recover useful functions since it can rely on the influence of the global variables $\mathbf{z}$ to contribute to the function prior irrespective of the influence of $k_{\text{aux}}$.

### 5.1 Synthetic classification example

We first illustrate the performance of the proposed model on a classification example. We generate a dataset of 100 data points and four classes, and use a BNN with one hidden layer of 50 hidden units with ReLU non-linearities, and two dimensional latent variables $\mathbf{z}$. Figure 4 shows the predictive performance of the proposed priors and various alternatives, including BNN (with unit Normal priors on weights) with mean field Gaussian variational approximation (MFVI) [Blundell et al., 2015] and

Hamiltonian Monte Carlo (HMC) [Neal, 1993], variational deep kernel learning (DKL) Wilson et al. [2016] and MetaNN [Karaletsos et al., 2018]. We highlight that *MetaGP-local* with RBF kernel gives uncertainty estimates that are reminiscent to that of a GP model in that the predictions express *"I don't know"* away from the training data, despite being a neural network under the hood. Following Bradshaw et al. [2017], we also show the uncertainty for data further from the training instances. *MetaGP-local*(RBF) remains uncertain, as expected, for these points while MFVI and DKL produce arguably overconfident predictions.

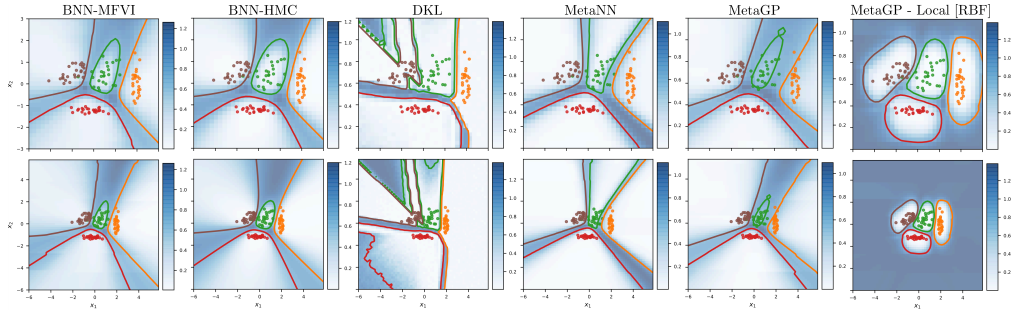

Figure 4: Predictive performance of various methods on a four-way classification problem. We compare the proposed approaches (MetaGP, MetaGP with an input-dependent RBF kernel and periodic kernel) to BNN with MFVI and HMC, DKL and MetaNN. Best viewed in colour. The background color shows the entropy of the predictive distribution. The contours show the 0.7 equiprobability contours. The bottom plots are the zoom-out version of the corresponding top plots, showing the predictive entropy further from the training points.

## 5.2 Inductive Biases For Neural Networks With Input-Dependent Kernels

We explore the utility of the input-dependent prior towards modeling inductive biases for neural networks and evaluate predictive performance on a regression example. In particular, we generate 100 training points from a synthetic sinusoidal function and create two test sets that contain in-sample inputs and out-of-sample inputs, respectively. We test an array of models and inference methods, including BNN (with unit Normal priors on weights) with MFVI and HMC, GPs with diverse kernel functions, DKL, MetaGP and local-MetaGP with input dependence given the same kernels as the GPs. We use RBF and periodic kernels [MacKay, 1998] for weight modulation and the pure GP in this example. Figure 5 summarizes the results. Note that the periodic kernel allows the BNN to discover and encode periodicity in its weights, leading to long-range confident predictions compared to that of the RBF kernel and significantly better extrapolation than BNNs with independent weight priors can obtain given the amount of training data, even when running HMC instead of VI.

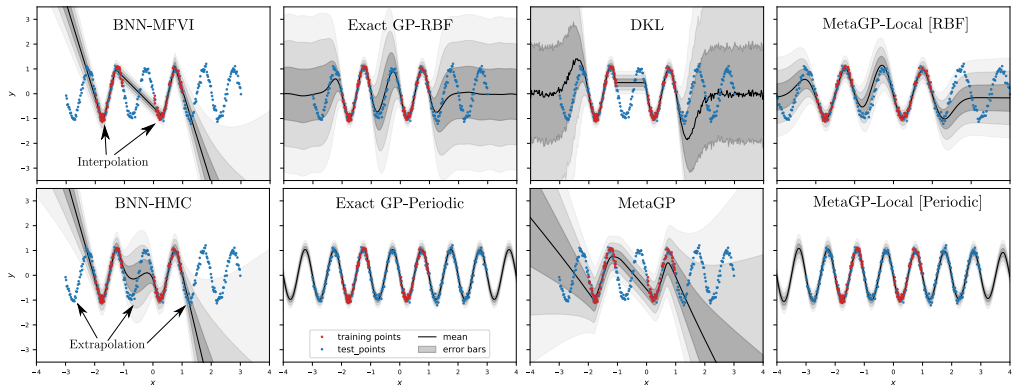

Figure 5: Predictive performance of various methods on a sinusoidal dataset. We also provide a quantitative comparison in Table 1.

We evaluate the quantitative utility of input-dependence and inductive biases on two test sets that contain in-sample inputs (between the training data) and out-of-sample inputs (outside the training

range), respectively. We report the performance of all methods in Table 1. The performance is measured by the root mean squared error (RMSE) and the negative log-likelihood (NLL) on the test set, and we evaluate separately for *extrapolation* and *interpolation*. In this example, the local MetaGP model is comparable to GP regression with a periodic kernel and superior to other methods, demonstrating good RMSE and NLL on both in-distribution and out-of-distribution examples.

Table 1: Avg. test error and NLL for the sinusoid example, averaged over five runs. Lower is better.

| Method | Interpolation | | Extrapolation | |
|---|---|---|---|---|
| | RMSE | NLL | RMSE | NLL |
| BNN-MFVI | $0.17 \pm 0.003$ | $-0.04 \pm 0.032$ | $3.51 \pm 0.170$ | $88.12 \pm 5.853$ |
| BNN-HMC | $0.12 \pm 0.000$ | $-0.69 \pm 0.000$ | $4.34 \pm 0.034$ | $10.98 \pm 2.725$ |
| Exact GP-RBF | $\mathbf{0.11 \pm 0.000}$ | $\mathbf{-0.81 \pm 0.000}$ | $0.55 \pm 0.005$ | $0.75 \pm 0.003$ |
| Exact GP-Periodic | $\mathbf{0.11 \pm 0.000}$ | $-0.80 \pm 0.001$ | $\mathbf{0.11 \pm 0.000}$ | $\mathbf{-0.83 \pm 0.000}$ |
| DKL | $0.12 \pm 0.005$ | $-0.72 \pm 0.047$ | $0.76 \pm 0.059$ | $3.26 \pm 1.885$ |
| MetaNN | $0.24 \pm 0.000$ | $0.26 \pm 0.000$ | $1.77 \pm 0.000$ | $12.14 \pm 0.000$ |
| MetaGP | $0.24 \pm 0.019$ | $0.08 \pm 0.068$ | $2.59 \pm 0.374$ | $5.86 \pm 0.577$ |
| MetaGP-Local[RBF] | $\mathbf{0.11 \pm 0.001}$ | $-0.80 \pm 0.014$ | $0.74 \pm 0.150$ | $1.50 \pm 0.200$ |
| MetaGP-Local[Periodic] | $\mathbf{0.11 \pm 0.002}$ | $-0.76 \pm 0.025$ | $0.12 \pm 0.003$ | $-0.69 \pm 0.039$ |

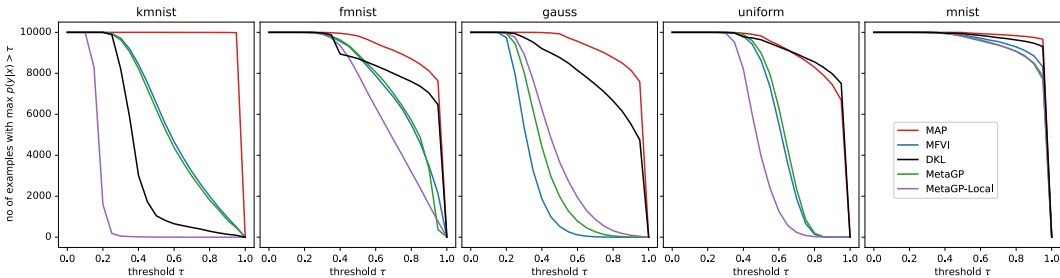

Figure 6: The number of samples with the highest predictive probability larger than a threshold for various methods. Following Snoek et al. [2019], we first obtain the confidence as $\max_k p(y = k|x, \mathcal{D})$. We filter out test examples corresponding to a particular confidence threshold $\tau \in [0, 1]$ and count those samples for each data set.

## 5.3 Input Dependent Neural Networks For Uncertainty Quantification

Motivated by the performance of the proposed *MetaGP-local* model in the synthetic examples in Figure 5, this section tests the ability of this model class to produce calibrated predictive uncertainty to out-of-distribution (OOD) samples. That is, for test samples that do not come from the same training distribution, a robust and well-calibrated model should produce uncertain predictive distribution and thus high predictive entropy. Such a model could find applications in safety-critical tasks or in an area where detecting unfamiliar inputs is crucial such as active learning or reinforcement learning. In this experiment, we train a BNN classifier with one hidden layer of 100 rectified linear units on the MNIST dataset, with *MetaGP-local*-RBF only applied to the last layer of the network. The dimensions of the latent variables and the auxiliary inputs are both 2, with auxiliary inputs given by transforming MNIST images using a jointly learned linear projection $\mathbf{V}$. After training on MNIST, we find the predictions on various test sets, including notMNIST, fashionMNIST, Kuzushiji-MNIST, and uniform and Gaussian noise inputs. Following [Lakshminarayanan et al., 2017, Louizos and Welling, 2017], the plots of the number of samples with preditive confidence larger than a threshold are shown in Fig. 6. A calibrated classifier should give a curve that bends towards the bottom-left corner of the plot for OOD examples and, vice versa, towards the top-right corner of the plot for in-distribution inputs. In most OOD datasets considered, except Gaussian random noise, *MetaGP* and *MetaGP-local* demonstrate superior performance to all comparators, including DKL. Notably, MAP estimation, often deployed in practice, tends to give wildly poor uncertainty estimates on OOD samples. We illustrate this behaviour and that of other methods on representative inputs of the Kuzushiji-MNIST and MNIST digits in the appendix. We also plot the PDFs and CDFs of the predictive entropies for various methods in the supplementary material.

# 6  Related work

There is a long history of research on developing (approximate) Bayesian inference methods for BNNs, i.e. in [Neal, 1993, 2012, Ghahramani, 2016]. Our work differs in that the model employs a hierarchical prior, and inference is done in a lower-dimensional latent space instead of the weight space. The variational approximation is chosen such that the marginal distribution over the weights is non-Gaussian and the correlations between weights are retained, in contrast to the popular mean-field Gaussian approximation. Imposing structure over the weights with a carefully chosen prior has been observed to improve predictive performance [Ghosh et al., 2018, Neal, 2012, Blundell et al., 2015], but it has remained elusive how to express prior knowledge or handle interpolation or extrapolation in such models. Modern deep Bayesian learning approaches often involve fusing neural networks and GPs, such as in deep kernel learning [Wilson et al., 2016], which layers a GP on top of a neural network feature extractor.

Another notable example is [Pearce et al., 2019], which blends kernels and activation functions to induce desired properties through architectural choices, but is not expressing these assumptions as a weight prior. The functional regularization approach introduced in [Sun et al., 2019] shares some of the motivations with our paper, but implements it very differently by explicitly instantiating a GP and performing a complex training scheme to learn neural networks that match that GP. Asymptotically, they match the GP, while in our model (i) the properties we care about are already built into the weight prior allowing direct training on a dataset without the involved minimax approach, and (ii) our posterior can depart from that restrictive prior as it fundamentally only guides a weight based model, i.e. by learning posterior kernel parameters for $k_{\mathrm{aux}}$ to eliminate its influence on $k_{\mathrm{local}}$ (such as wide lengthscales).

When comparing our local model directly to a GP, the representational differences are also stark: a GP can only learn functions from the space defined by the kernel, whereas our model can depart from the auxiliary kernel used for input dependence by virtue of the global variables that can generate weights. We illustrate an example of this in Sec. D Fig. 8, where we show that our model can function under misspecification, where a GP may fail. We note that methods such as kernel search [Duvenaud et al., 2013] or spectral mixture kernels [Wilson and Adams, 2013] can also be used to 'learn' appropriate kernels for GPs, and could also be useful in combination with our local model.

David J.C. MacKay famously poses the question *"Have we thrown the baby out with the bathwater?"* when discussing GPs and neural networks [MacKay, 2003]. We believe that by combining ideas from both fields we can posit models with representational abilities combining the benefits of each, while maintaining a weight-based representation.

Another related theme is hyper-networks, the core idea of which is to generate network parameters using another network [see e.g. Ha et al., 2016, Stanley et al., 2009]. Our model resembles a hyper-GP based on a GP-LVM [Lawrence, 2004], with a key structural assumption of node latent variables as introduced in [Karaletsos et al., 2018] to enable compact prediction per weight instead of per weight tensor.

# 7  Summary

We proposed a GP-based hierarchical prior over neural network weights, a variant permitting input-dependent weight priors, and an effective approximate inference strategy. A key feature of the input-dependent priors is the fact that they are local to each datapoint, or indeed each layer, depending on application. We demonstrate utility of these models for interpolation, extrapolation and uncertainty quantification, outperforming strong baselines and showing the utility of the input-dependent weight prior, which combines strengths of both GPs and NNs. We see that compared to N(0,1) weight priors our models have qualitatively and quantitatively different behaviors in generalization tasks, even when compared against HMC-inferred networks, suggesting that the study of per-datapoint priors is an interesting avenue of investigation in addition to inference for BNNs. We plan to evaluate the performance of the model on more challenging decision making tasks and to study more elaborate conditioning mechanisms in the context of specific applications.

## Broader Impact

Our work targets studying priors of neural networks with respect to two specific aspects: first, we aim at obtaining weights which are sharp close to the training data and uncertain away from training data, in order to calibrate the model's confidence. This is essential for many applications where predictions of neural networks are consumed to drive decisions, which may occur a cost. In case our model produces "I don't know" predictions as we showed it is capable of in OOD data, ML-systems can either probe an expert or utilize a fallback plan for decisions. Such cases occur across industrial applications of algorithmic decision making and impact economics and fairness, but are even more critical in fields such as healthcare or autonomy where wrong but overconfident predictions may lead to catastrophic decisions.

The second area of impact centers around the ability of a practitioner to express specific types of prior knowledge for the functions learned by a neural network via auxiliary kernels. This can help practitioners utilize neural networks as less of a black box and ultimately may lead to the ability to train networks with rich weight-based function spaces with little data. These types of network regularization are application-dependent, but ultimately we hope structures such as the ones we propose may be able to aid with generalization outside the training data by encoding prior knowledge into networks, an ability that would potentially help in a variety of real world scenarios where data paucity exists but prior knowledge can be used to fill the gaps.

## Acknowledgments and Disclosure of Funding

We would like to thank Christian Perez and the anonymous reviewers for the comments. The authors did not receive any third party funding or third party support, or have financial relationships with entities that could potentially be perceived to influence what was written in the submitted work, during the last 36 months prior to this submission.

## Footnotes

[2]Here, we present a homoscedastic noise model for the weights, but the model is readily adaptable to a heteroscedastic noise model which we omit for clarity.

[3]The conditioning on $\mathcal{C}_{\neq \mathbf{C_u}}$ and $\mathbf{C_u}$ in $p(\mathbf{u})$ and $p(f_{\neq \mathbf{u}} | \mathbf{u})$ is made implicit here and in the rest of this paper.

[4]We can also use the *local reparameterization trick* [Kingma et al., 2015] to reduce variance.

[5]Additional updates and results will be available on `https://arxiv.org/abs/2002.04033`.

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
