[Supplementary Material]

# A   Additional Background on Bayesian neural networks and variational inference

Consider a training set comprising of $N$ input-output pairs, $\mathcal{D} = \{\mathbf{x}_n, y_n\}_{n=1}^N$, and a neural network parameterized by weights and biases, $\mathbf{w}$, that describes the distribution over an output $y_n$ given an input $\mathbf{x}_n$, $p(y_n|\mathbf{w}, \mathbf{x}_n)$. We follow a Bayesian approach by placing a prior distribution over the network parameters, $p(\mathbf{w})$, and obtaining the posterior distribution $p(\mathbf{w}|\mathcal{D})$, which involves calulation of the marginal likelihood $p(\mathcal{D}) = \int d\mathbf{w} p(\mathbf{w}) p(\mathcal{D}|\mathbf{w})$. However, obtaining $p(\mathbf{w}|\mathcal{D})$ and $p(\mathcal{D})$ exactly is intractable when $N$ is large or when the network is large and as such, approximation methods are often required. In particular, mean-field Gaussian variational inference (MFVI) has recently become a method of choice for approximate inference for Bayesian neural networks due to its simplicity and the recently popularized *reparameterization trick* [Salimans and Knowles, 2013, Kingma and Welling, 2013, Titsias and Lázaro-Gredilla, 2014, Blundell et al., 2015]. MFVI sidesteps the intractability by positing a diagonal Gaussian approximation $q(\mathbf{w}) = \mathcal{N}(\mathbf{w}; \boldsymbol{\mu}, \mathrm{diag}(\boldsymbol{\sigma}^2))$ and optimising an approximate lower bound to the marginal likelihood $\mathcal{L}_{\mathrm{MFVI}}(q(\mathbf{w})) \approx -\mathrm{KL}[q(\mathbf{w})||p(\mathbf{w})] + \frac{1}{K}\sum_{k=1}^{K}\sum_{n=1}^{N}\log p(y_n|\mathbf{w}_k, \mathbf{x}_n)$, where $\mathbf{w}_k = \boldsymbol{\mu} + \boldsymbol{\sigma} \odot \epsilon_k$ and $\epsilon_k \sim \mathcal{N}(\mathbf{0}, \mathbf{I})$, i.e. $\mathbf{w}_k$ is a sample from $q(\mathbf{w})$. Note that the mean-field variational Gaussian approximation with a standard normal prior, presented in is often outperformed by point estimation in certain settings [Trippe and Turner, 2018]. Despite being practical and able to give reasonable uncertainty estimates, improving MFVI is still an active research area, and the main focuses of which are (i) improving the reparameterization gradient estimator to enable faster convergence [Miller et al., 2017, Wu et al., 2018], (ii) replacing the typical standard Normal prior, $p(\mathbf{w}) = \mathcal{N}(\mathbf{w}; \mathbf{0}, \mathbf{I})$ by a structured prior that better models the structures present in the weight a-priori [Ghosh et al., 2018, Neal, 2012, Blundell et al., 2015], and (iii) using structured variational approximations that can potentially capture weight correlations in the posterior [Louizos and Welling, 2016, Zhang et al., 2017]. This paper builds on the two latter themes and proposes a hierarchical model for the prior and a structured variational scheme that explicitly model and infer weight structures.

# B Illustration of variants of the proposed model

Figure 7 shows different variants of the proposed prior and how different input signals can be used for the local kernel.

Figure 7: Illustration of different variants of the proposed model: (a) a network where the a single latent function is used to generate weights for all layers, (b) each layer can have its own latent function, (c) and (d) the prior can be made input-dependent by using the auxiliary kernel and can be combined with (a) or (b), respectively, (e) the auxiliary kernel can use some other signals, for example, the output of the previous layer.

## B.1 Exploring the effect of parametrizations on the capacity and expressivity of GP-MetaPriors

Here, we shows different variants of MetaGP-local with varying parametrizations for the Gaussian Process in relation to scalability and expressivity.

**Global-aux** is the simplest model, where inducing points for auxiliary kernel $k_{aux}$ and weight kernel $k_w$ are shared across the entire network. As such, the parameters here correspond to the numbers of inducing points.

**layer-aux** is a model where each weight layer has its own kernel parameters and inducing points for both $k_{aux}$ and $k_w$. This model shows how we can effecrtively scale to arbitrary neural networks by having compact parametrizations per layer in terms of inducing points and by modeling the influence of the input (say, the input variable x) on the network via the auxiliary kernel in a per-layer-fashion.

The total number of parameters here correspond to the numbers of inducing points (and kernel parameters) times the number of weight layers.

**layer-stacked-aux** is a model where each weight layer has its own kernel parameters and inducing points for both $k_{aux}$ and $k_w$, but unlike **layer-aux** the auxiliary kernel is not conditioned on the input $x$ at each layer, but on the output of the previous layer (and $x$ on the bottom layer). This model can be understood as a variant that stacks the local model we introduce per layer and may make sense in application with auxiliary kernels that will promote uncertainty, but is less suitable towards regularizing function space with informative kernels, as it is hard to design informative kernels for intermediate layer responses. We intend to study this variant in more detail in future work, but add it here for illustrative purposes.

We test the models on two neural networks for a regression task on the Boston UCI-dataset with $[50]$ hidden units and $[50, 50]$ in the single layer and two-layer local models respectively. We also vary the amount of inducing points we afford each kernel. As auxiliary kernels we only use RBF kernels here. We learn an output noise $\sigma^2$ for homoschedastic regression via maximum likelihood. The experimental results in Table 2 suggest that a layer-wise parametrization has benefits over a global parametrization and that adding inducing points can increase performance up to saturation. In this case, we see that the single layer models do not benefit from more inducing points, likely due to the small dataset size. We also see that the stackable version of the model performs best with the fewest inducing points in this small data setting.

Table 2: Average test error (RMSE) and test negative log-likelihood (NLL) for the Boston dataset (with 13 input dim) are shown, for various choices of kernel parametrizations, network sizes, and amounts of inducing points per kernel.

| Method | RMSE | NLL | RMSE | NLL | RMSE | NLL |
|---|---|---|---|---|---|---|
| | 10 I.P. | | 20 I.P. | | 40 I.P. | |
| 1l-global-aux | 3.93. | -2.76. | 3.90. | -2.86 | 4.05 | -2.98 |
| 1l-layer-aux | **3.47** | -2.72 | 3.91 | -2.96 | 3.84 | -3.32 |
| 2l-global-aux | 5.28 | -3.19 | 3.87 | -3.09 | 4.078 | -2.88 |
| 2l-layer-aux | 4.22 | -2.99 | 3.48 | **-2.56** | 3.34 | -2.60 |
| 2l-layer-stacked-aux | **3.31** | **-2.56** | 3.94 | -2.64 | 3.37 | -2.54 |

# C   Inference for the local model

The main difference in the local model is the dependence of weights on inputs. To handle inducing point kernels over both weight codes and inputs, we introduce inducing inputs $\tilde{\boldsymbol{\epsilon}} = \{\tilde{\epsilon}_m\}_{m=1}^M$ where $\tilde{\epsilon} \in \mathcal{R}^{D_{\text{aux}}}$ for $k_{\text{aux}}$. We then concatenate the dimensions of $\mathbf{C_u}$ in Section 4 with the new inducing inputs to form the new inputs $\tilde{\mathbf{C}}_\mathbf{u} = [\mathbf{C_u}; \tilde{\boldsymbol{\epsilon}}]$. The set of inputs $\tilde{\mathbf{C}}_\mathbf{u}$ now have dimensions $\tilde{\mathbf{c}}_u \in \mathcal{R}^{2D_z + D_{\text{aux}}}$.

The fully instantiated covariance matrix $\mathbf{K}_{\text{local}} = K_w \otimes K_{\text{aux}}$ would take the shape $|\mathbf{w}| \times |\mathbf{w}| \times N \times N$. As this kernel has Kronecker structure one could now consider using inference techniques such as in Flaxman et al. [2015]. However, the tractability of the global kernel remains an issue even in this case. As such, we elect to inherit the diagonal approximation from Section 4 and apply it to the joint kernel, yielding an object of dimension $|\mathbf{w}| \times N$. The lower bound computation in Section 4 can thus be reused but with $\mathbf{A}^{(n,s)}$ and $\mathbf{B}^{(n,s)}$ being input-dependent[6].

We can handle large datasets by using inducing point kernels, which permit inference using mini-batches. Another difference is the potential existence of the mapping $\mathbf{V}$ in the model, which we tackle by introducing a variational distribution $q(\mathbf{V}) = \mathcal{N}(\mathbf{V}; \boldsymbol{\mu}_\mathbf{V}, \text{diag}(\boldsymbol{\sigma}_\mathbf{V}^2))$. We can estimate the evidence lower bound by also drawing unbiased samples from this and jointly optimizing its parameters with the rest of the variational parameters. The overall computational complexity with data-subsampling in this section and Section 4 is $\mathcal{O}(M^3 + |\mathbf{w}|M^2)$.

# D Comparing BNNs with a MetaGP prior and exact GPs

Here, we study how the local vewrsion of our model compares to GP-regression. In a thought-experiment, we can easily show that our model can learn to ignore the auxiliary kernel $k_{aux}$ and learn purely weight-based functions that generalize outside of that part of its prior, as it also has access to latent variables $\mathbf{z}$ to specify functions. In the example shown in Fig. 8, we use the 'airline' dataset and study how a misspecification in kernel space will impact the functions that are learned. This data contains more than a periodicity, as there is also a time-dependent trend to capture. What we observe is that a GP with a periodic kernel fails to model this function well, but once combined in a more complex kernel the GP can fit the data fine. However, this may require knowledge of the kernel, kernel learning, or kernel search (such as in [Duvenaud et al., 2013]). In contrast, when using a periodic kernel for our local model, the model learns a function that is outside of the function space defined by the periodic kernel $k_{aux}$ alone. This provides evidence that our model differs from a GP, and poses a qualitatively and quantitatively richer function prior.

Figure 8: Top and Middle: Predictions made by a BNN-MetaGP prior with a local periodic kernel and exact GP regression with a periodic kernel, respectively. Bottom: Predictions by exact GP regression with a (SE x Periodic + Linear) kernel.

# E   Prior samples from the global model

We show prior samples from this model in Fig. 9 by the following procedure: for a sample $\mathbf{z}_{\text{viz}} \sim p(\mathbf{z})$ we instantiate the covariance matrix $\mathbf{K}_w$ by constructing weight codes and applying the kernel function $k_w$. We draw weights from the GP by sampling the Normal distribution $\mathcal{N}(\mathbf{w}; \mathbf{0}, \mathbf{K}_w + \sigma_w^2 \mathbf{I})$, where $\mathbf{K}_w \in \mathcal{R}^{|\mathbf{w}| \times |\mathbf{w}|}$, with $|\mathbf{w}|$ denoting the number of all parameters in the network. We then generate BNN function samples given the sampled weights with homoscedastic noise on the outputs. We highlight two properties of this model: *First*, as a hierarchical model, given a sample of the latent variables $\mathbf{z}$, the model instantiates a prior over weights from which we can further sample functions and thus encodes two levels of uncertainty over functions. *Second*, we demonstrate that changing the length-scale parameter of the mapping-kernel $k_w$, again even *given fixed samples* $\mathbf{z}$, leads to vastly differentiated function samples, showing the compact degree of control the mapping parameters have over the function space being modeled. In this case, the length-scale appears to control the variance over the function space, which matches an intuitive interpretation over the kernel parameter.

Figure 9: **MetaGP Prior Samples**: We show function samples generated from a [1,20,10,1] unit BNN with ReLUs with meta-GP prior. We draw one sample per unit $\mathbf{z}_{\text{viz}} \sim p(\mathbf{z})$ to instantiate the weight prior and subsequently draw 40 function samples (individually colored) from the BNN by drawing from the conditional prior $\mathbf{w} \sim \mathcal{N}(\mathbf{w}|\mathbf{z}_{\text{viz}})$ and regressing $\mathbf{y} \sim p(\mathbf{y}|\mathbf{w}, \mathbf{x})$. (**left**) We show samples drawn when the global RBF kernel for the GP has a length scale set to a small value. (**middle**) we keep the same samples $\mathbf{z}$ and only change the length scale to be larger and visualize the functions induced by the BNN (**right**) We visualize the weight kernel given the latent variables $\mathbf{z}_{\text{viz}}$. Other samples of $\mathbf{z}$ would induce different weight covariance matrices. Overall this figure shows that even given $\mathbf{z}$, the proposed prior models a wide range of functions which have controllable properties based on the parameters of the kernel.

# F  Prior samples from the local model

We demonstrate the effects of utilizing the auxiliary kernel in this factorized fashion by visualizing prior function samples from a BNN with this local prior when changing kernel parameters in Fig. 10, exemplifying the proposed model's ability to encode controlled periodic structure into BNN weight priors before seeing any data. As performed in Section 3, we sample from the GP to instantiate weights, but in the case of the local model we instantiate the covariance matrix $\mathbf{K}_{\mathrm{local}} \in \mathcal{R}^{|\mathbf{w}| \times |\mathbf{w}| \times N \times N}$. We discuss the handling of this conceptually large object in Appendix C.

Figure 10: **Local MetaGP Samples**: We remind the reader that the input-dependent weight prior has a factorized kernel structure $k_{\mathrm{local}} = k_w \cdot k_{\mathrm{aux}}$, and we wish to demonstrate the effect of each kernel separately in terms of its effects on the induced function prior for the neural network. We are given the same samples $\mathbf{z}_{\mathrm{viz}}$ as in Fig. 2 and also keep the two kernel parameter choices for $k_w$, while varying only the period parameter for an auxiliary periodic kernel. **Left**: We show function samples using a small period of 0.1 and a period of 0.3 in combination with the $k_w$ kernel with length-scale 0.8. We can see, that while the functions are still relatively flat, the auxiliary kernel induces weight priors which lead to periodic function samples consistent with the auxiliary kernel setting. **Right**: Similarly, when performing the same protocol for the $k_w$ with the larger lengthscale, we again observe periodic functions consistent with the set period (although we only apply the periodic kernel for the weight priors for the BNN), but see that the functions sampled have more variance, consistent with the larger length-scale of the weight-kernel $k_w$. Note that while the functions exhibit periodic structure, they have non-periodic global structure as well, as they also draw information from $k_w$ and the priors are merely *modulated* by the auxiliary kernel. We thus see that our prior structure successfully induces function priors which naturally inherit properties we can express as kernel functions, but keep rich expressivity as weight based models.

# G Weight structures

The marginal and conditional covariances a small neural network under the proposed prior are shown in Fig. 11.

Figure 11: Marginal and conditional covariance structures over weights in a 1x50x1 BNN. Sampling from the posterior of the hierarchical model reveals that even a diagonal GP approximation can capture off-diagonal correlations induced through unit correlations. Also note the off-diagonal bands in the marginal plots above, which indicate the correlation structures induced by the latent variables of the hidden units connecting the layers. We remove the diagonal in the marginal plots for clarity.

# H  Extra experimental results

## H.1  Details of the networks and models used in the experiments

We detail the network and training configuration for the experiments shown in the main text in Tables 3 to 5.

Table 3: Configurations for the synthetic classification experiment in sec 5.1

| Configuration | Value |
|---|---|
| Hidden layer | [50] |
| Dimension of latent variable $z$ | 2 |
| Number of inducing points | 50 |
| Number of training points | 100 |
| Number of MC samples for training | 2 |
| Learning rate | 0.005 |

Table 4: Configurations for the experiment with the sinusoid dataset in sec 5.2

| Configuration | Value |
|---|---|
| Hidden layer | [50] |
| Dimension of latent variable $z$ | 2 |
| Number of inducing points | 50 |
| Number of training points | 100 |
| Number of MC samples for training | 2 |
| Learning rate | 0.005 |

Table 5: Configurations for the MNIST classification experiment in sec 5.3

| Configuration | Value |
|---|---|
| Hidden layer | [50] |
| Dimension of latent variable $z$ | 1 |
| Dimension of $\epsilon$ for local model | 2 |
| Number of inducing points | 50 |
| Number of training points | 60000 |
| Number of MC samples for training | 2 |
| Learning rate | 0.005 |
| Number of epochs | 500 |
| Batch size | 200 |

## H.2 MNIST experiment: full figures

In this section, we include the full figures of the MNIST out-of-distribution uncertainty experiment. The full results of all models/methods considered are shown in Figs. 12 to 14. These metrics were suggested by Snoek et al. [2019]. MetaGP with the input-dependent kernel shows good performance, outperforming all other methods in almost all cases. In addition, we include the full figures for the predictive distributions on representative test examples in Figs. 15 and 16.

Figure 12: PDF of predictive entropies on MNIST and non-MNIST test points. Best viewed in colour.

Figure 13: CDF of predictive entropies on MNIST and non-MNIST test points. Best viewed in colour.

Figure 14: Count of test samples with maximum class probability larger than a threshold. Best viewed in colour.

Figure 15: Predictive distribution for representative MNIST test examples by various methods. Best viewed in colour.

Figure 16: Predictive distribution for representative KMNIST test examples by various methods. Best viewed in colour.

## H.3 An empirical evaluation of various approximations for $q(\mathbf{w}|\mathbf{z}_k, \mathbf{V}_k, \mathbf{x})$

In this section, we analyze the impact of different approximations to the covariance matrix of $q(\mathbf{w}|\mathbf{z}_k, \mathbf{V}_k, \mathbf{x})$:

$$q(\mathbf{w}|\mathbf{z}_k, \mathbf{V}_k, \mathbf{x}) = \mathcal{N}(\mathbf{w}; \mathbf{A}^{(k)}\boldsymbol{\mu}_u, \mathbf{B}^{(k)} + \mathbf{A}^{(k)}\Sigma_{\mathbf{u}}\mathbf{A}^{\mathsf{T},(k)}).$$

If we use the exact, fully correlated Gaussian distribution above, it is necessary to sample from this distribution to evaluate the lower bound. This step costs $\mathcal{O}(W^3)$ where $W$ is the number of parameters in the network.

The complexity can be greatly improved by making a diagonal approximation to $\mathbf{B}^{(k)}$ as follows,

$$\hat{q}_{\mathrm{FITC}}(\mathbf{w}|\mathbf{z}_k, \mathbf{V}_k, \mathbf{x}) = \mathcal{N}(\mathbf{w}; \mathbf{A}^{(k)}\boldsymbol{\mu}_u, \mathrm{diag}(\mathbf{B}^{(k)}) + \mathbf{A}^{(k)}\Sigma_{\mathbf{u}}\mathbf{A}^{\mathsf{T},(k)}).$$

Sampling from this distribution can be done in $\mathcal{O}(WM^2)$ where M is the number of pseudo-points.

This can be further approximated by assuming a diagonal covariance matrix,

$$\hat{q}_{\mathrm{diag}}(\mathbf{w}|\mathbf{z}_k, \mathbf{V}_k, \mathbf{x}) = \mathcal{N}(\mathbf{w}; \mathbf{A}^{(k)}\boldsymbol{\mu}_u, \mathrm{diag}(\mathbf{B}^{(k)} + \mathbf{A}^{(k)}\Sigma_{\mathbf{u}}\mathbf{A}^{\mathsf{T},(k)})).$$

The variational bound can then be evaluated by drawing samples from $\hat{q}_{\mathrm{diag}}$ as in the above approximation, or by drawing activity samples by employing the local reparameterization trick [Kingma et al., 2015].

We evaluate the performance of using the exact and approximate conditional distributions above in a range of toy regression and classification, and show representative results in Figs. 17 and 18. We note that the diagonal approximation is fast and gives qualitatively similar performance compared to more structured approximation or the exact case, in both cases where there is a single GP for all weights in the network and there is multiple GPs, one for each weight layer in the network.

Figure 17: An evaluation of the covariance matrix approximations in a toy regression example. Top: objective function during training vs epoch/time. Bottom: Predictions after training using one of the approximations discussed in the text. Global: there is one GP for all weights in the network. Layer: there are multiple GPs, one for each weight layer in the network. Note that we are not using the auxiliary kernel here. Best viewed in colour.

Figure 18: An evaluation of the covariance matrix approximations in a toy classification example. Top: objective function during training vs epoch/time. Bottom: Predictions after training using one of the approximations discussed in the text. Global: there is one GP for all weights in the network. Layer: there are multiple GPs, one for each weight layer in the network. Note that we are not using the auxiliary kernel here. Best viewed in colour.

## H.4 Results on a synthetic regression example

In this section, we demonstrated the performance of the proposed priors on a 1D test function, as used in [Louizos et al., 2019]. We compare to BNN with independent Gaussian priors and a mean-field Gaussian variational approximation, and MetaNN [Karaletsos et al., 2018]. The training points, and predictive mean and error bars are shown in Fig. 19.

Figure 19: Predictive performance of various methods on a 1D test function. We compare the proposed approaches (MetaGP and MetaGP with an input-dependent kernel) to BNN-MFVI and MetaNN. Best viewed in colour.

## H.5 Robustness in various data regimes for a toy regression problem

In this experiment, we evaluate the qualitative performance of various methods, including MFVI, MetaNN, MetaGP and MetaGP with local, input-dependent kernel, on a toy regression problem, in different data regimes. In particular, we considers 10, 20, and 50 training points respectively, and plot the predictions in Fig. 20. MetaGP demonstrates consistent performance across all data regimes, comparable to that of MetaNN. The input-dependent kernel helps the performance further in the out-of-distribution area.

Figure 20: Performance of mean-field variational inference, MetaNN with variational inference and MetaGP with variational inference on a toy regression problem with various number of training points. Best viewed in colour.

## H.6 Robustness of MetaGP with network architectures

In this experiment, we compare the performance of MetaGP for various numbers of hidden units (20, 50 and 100) and two activation functions (Tanh and ReLU) on a toy regression problem. The observation noise is fixed in this experiment. We observe that the performance of the models is in general consistent across different activation functions and numbers of hidden units. We show the results in Fig. 21.

Figure 21: Performance of MetaGP on a toy regression problem, with various numbers of hidden units and different activation functions. Best viewed in colour.

## H.7   Effect of input-dependent kernels

To understand the impact of the auxiliary kernel to the prediction, we use a model trained on the sinusoid dataset, as shown in the main text, and vary the period hyper-parameter in the kernel whilst keeping other hyper-parameters and variational parameters fixed. The predictions for a few hyperparameters are shown in Fig. 22. We note the variation/period in the data is captured by weight modulation, governed by the input-dependent kernel. Changing the period hyperparameter affects how fast or slow the weights are changing wrt the input.

Figure 22: We first train a model with an input-dependent kernel on a sinusoid data set (top left) and then vary the period hyperparameter of the input-dependent kernel whilst keeping other hyperparameters and variational parametes fixed (others). Best viewed in colour.

## Footnotes

[6]Specifically, $\mathbf{A}^{(n,s)} = \mathbf{K}_{\mathbf{wu}}^{(n,s)}\mathbf{K}_{\mathbf{uu}}^{-1}, \mathbf{B}_n^{(n,s)} = \mathbf{K}_{\mathbf{ww}}^{(n,s)} - \mathbf{K}_{\mathbf{wu}}^{(n,s)}\mathbf{K}_{\mathbf{uu}}^{-1}\mathbf{K}_{\mathbf{uw}}^{(n,s)} + \sigma_w^2\mathbf{I}$, where $\mathbf{K}_{\mathbf{wu}}^{(n,s)} = k_w(\mathbf{C}_\mathbf{w}^{(s)}, \mathbf{C_u}) \otimes k_{\text{aux}}(\mathbf{x}_n, \tilde{\boldsymbol{\epsilon}}), \mathbf{K}_{\mathbf{uu}} = k_w(\mathbf{C_u}, \mathbf{C_u}) \cdot k_{\text{aux}}(\tilde{\boldsymbol{\epsilon}}, \tilde{\boldsymbol{\epsilon}}), \mathbf{K}_{\mathbf{ww}}^{(n,s)} = k_w(\mathbf{C}_\mathbf{w}^{(s)}, \mathbf{C}_\mathbf{w}^{(s)}) \otimes k_{\text{aux}}(\mathbf{x}_n, \mathbf{x}_n)$.