[Reviews · NeurIPS 2020]

Review 1

Summary and Contributions: The authors investigate structured priors over the weights of a Bayesian neural networks and propose the use of a Gaussian process to capture correlations among network weights. The construction involves endowing each unit of the neural network with latent variables and defining the weights between two units as a function of their respective latent variables. This function is modeled as a Gaussian process (GP). The authors also introduce a “local” variant, where the weights are a function of both the latent variables and the observed covariates. The local variant is attractive, as it allows users to specify desired inductive biases over functions (just like one would when GPs are used to directly model the functional mappings between covariates and responses).

Strengths: I like many of the ideas presented in this paper. * The notion of allowing for a-priori correlations among network weights is sensible, and using GPs to model such correlations seems natural. * Endowing the units of the network with latent variables and defining the network weights as a function of these, is another neat idea that allows for more tractable inference (latent variables are per unit, hence lower dimensional than the weights) and potentially allows one to bake in interesting inductive biases. * The experiments do a reasonable job of comparing various BNN variants, but see comment below.

Weaknesses: **** Post Rebuttal **** The rebuttal addressed my concerns. I am raising my score from 5 to 6. Although concerns with scalability remain, I think the ideas in the paper are interesting. The paper would be greatly strengthened by including illustrative scenarios where BNNs with local MetaGP priors model the data better than a GP with an identical kernel. In the rebuttal the authors claimed that the local MetaGP models could learn to ignore k_aux. Illustrative examples demonstrating this (I imagine this would happen when k_aux is misspecified — for example, data is not periodic but k_aux is chosen to be periodic) would be interesting to include in the paper. ************* My main concern can be summarized in one sentence: Why not just use a GP to model the data? What does the proposed construction buy us over GPs? * Scaling issues: The scaling issues associated with GPs only get worse in the proposed construction. The function drawn from the GP now needs to be evaluated at both the data and for each pair of units connected by a weight in the model for MetaGP-Local. Things are somewhat better for the global MetaGP, but even here for moderate sized networks, the number of weights can be greater than the number of training instances. * Modeling benefits: Unclear if there is any added flexibility being afforded by the somewhat labored construction, GP -> Weights of a NN -> functions over data. It is possible that the inductive biases built into the architecture of the network are important for certain types of tasks, but the authors only demonstrate results with fully connected architectures, with no specific biases. * Experiments: Only one toy experiment includes comparisons against GPs, and the takeaway is that the proposed BNNs perform similarly to GPs. It would be interesting to see comparisons to GPs in the ood detection task in 5.3 Given these issues, I am leaning towards a reject, but could be convinced otherwise if the authors are able to point out why their BNN variant should be preferred to GPs.

Correctness: yes

Clarity: The paper is clearly written, and easy to follow.

Relation to Prior Work: The work is well placed in the context of previous work.

Reproducibility: Yes

Additional Feedback:


Review 2

Summary and Contributions: The paper presents a new way of defining priors for Bayesian neural networks. The idea is as follows: for each node in the network, define a latent variable (not to be confused with the activation of the node). Then prior mean for a weight connecting node i in layer l to node j in layer l+1 is obtained by mapping the concatenation of the latent variables corresponding to these nodes with a function f. Previously, a neural network has been used for f. The contribution of this paper is that f is taken to be a probabilistic model, a Gaussian process, resulting in what is termed the “GP-MetaPrior”. Another novelty is to make the prior “local”, i.e., specific to a given data point. This is achieved by giving the function f as input not only the node-specific latent variables (which are common to all data points) but also features of a specific datapoint, resulting in “Local GP-MetaPrior”. The benefit of the GP-MetaPrior is that the number of latent variables (equal to the number of nodes) is much smaller than the number of weights in the network. Thus, it is easier to model the LVs than the weights (though notably this benefit is already present in previous work that used the NN instead of GP to model the weights). Motivation for the local prior is that it causes the weights to revert to the prior for out-of-distribution (OOD) data samples, better reflecting the high uncertainty related to such samples, which previous methods have not been able to capture adequately. Furthermore, it’s possible to encode for example periodicity into the BNN prior by selecting a suitable periodic kernel in the local prior. The benefits of the method are demonstrated quite thoroughly using both synthetic and real-world data sets in interpolation, extrapolation, uncertainty quantification, and active learning.

Strengths: The paper addresses an important open question in the Bayesian neural network literature, the inability of the previous BNN methods to properly reflect uncertainty for OOD data points. The paper presents a novel approach that seems to provide a significant improvement compared to previous BNN priors in this respect. I found the ideas interesting and I believe the paper will be of interest to researchers working with the topics of uncertainty and BNNs. The presentations is very clear and technically correct, and the experiments thoroughly demonstrate the properties of the new method and its strengths compared to other methods (also in the extensive supplementary).

Weaknesses: Ensemble methods have recently emerged as a competitive approach for uncertainty quantification. Including those in the comparison would have strengthened the paper. At least they should be mentioned, with a clarification for why a comparison was not considered relevant. The neural networks considered are relatively small-scale, e.g. a single hidden layer with 50 nodes. Now that the goal of imposing GP-like properties to BNN priors has become popular in the field (in addition to this paper e.g. that of Sun et al. 2019), the reader is left to wonder why not just use the GPs in the first place? This could warrant some discussion. One benefit of NNs over GPs, as I understand it, is the better scalability in terms of the number of samples. Related to that, could the authors comment on how does the computational complexity of the Local GP-MetaPrior compares to that of the regular GP (as both anyway involve a kernel between data items)? Do the authors see any other advantages compared to (sparse) GPs? VERY MINOR: Fig. 2 right is just a blue square when printed. Please revise. Fig. 3 caption: $k_aux$ should be $k_{aux}$

Correctness: Yes.

Clarity: The paper is very clearly written and the properties of the method are intuitively demonstrated.

Relation to Prior Work: Yes.

Reproducibility: Yes

Additional Feedback: POST-REBUTTAL: I find the rebuttal did a good job in addressing one of the main concerns, the relation of the current method to GPs. I still hold my overall positive view on this paper (score 7). If accepted, I encourage the authors to improve the final version as they described in their rebuttal.


Review 3

Summary and Contributions: This paper proposes another layer of Bayesian treatment for BNN via replacing the NN parameterization for the conditional distribution of network weights given latent variables by a GP prior. With various practical considerations, inference/learning of the proposed method can be achieved with variational re-parameterization.

Strengths: I believe the main contribution of this paper is enabling uncertainty characterization for the BNN weights while being economical with the number of parameters. In my opinion, the approach proposed is relatively fresh and novel.

Weaknesses: Regarding the theoretical approach: I like the idea of imposing a GP prior to capture a probabilistic parameterization of p(w| f, c). However, as pointed out in the paper, full GP definitely has some complexity issue as the size of the kernel blows up to (|w|N)^2 with the auxiliary component to account for inputs. The paper then proposes a diagonal approximation to alleviate this. I'm not sure if this makes sense because it seems to suggest that for some data point xj, f([z_{l, u}, z_{l+1,v}, xj]) and f([z_{l', u'}, z_{l'+1,v'}, xj) are independent for all distinct pair (l, u, v) and (l', u', v'). Does this approximation scheme contradict the approach's intent to capture weight correlations in the prior? I would be interested to compare this with a block diagonal approximation which yield a kernel of size |w| x |w| x N, which only assumes independence of f across inputs and hopefully is still tractable for small networks. Regarding the practicality of this approach (which is my primary concern), even with the diagonal approximation, this approach would be too costly when |w| is large. I notice that all the experiments are conducted on shallow networks. At what point would all this gain from the Bayesian treatment be eclipsed by a deep network with sufficiently many hidden units? A fair comparison would be allowing the same training budget (cpu/gpu time) and benchmark the proposed BNN against regular deep network of varying sizes. Minor: Regarding Fig. 15, it seems to me that DKL is the best performing method here. Fig. 16 is a good demonstration of the proposed method being able to capture uncertainty. Can we see a repeat of Fig. 16 for notMNIST and fashionMNIST?

Correctness: The claims are correct. The collection of results is impressive, although I would like to see larger scale experiments.

Clarity: The paper is well written. I am able to understand every point.

Relation to Prior Work: Prior works are clearly discussed.

Reproducibility: Yes

Additional Feedback: Post rebuttal feedback: My main problem with this paper is the scalability issue. I do think that this paper is pretty far from being practical, seeing that in the rebuttal the authors said complexity per layer is O(WM^2) and in the supplementary they used M=50 so its at least a thousand time more expensive than a DNN with similar architecture. However, I acknowledge that this paper has many fresh ideas that could potentially be meaningful to the community. I will keep my score as 6.


Review 4

Summary and Contributions: The paper proposes GP-induced weights for BNNs, and input-dependent, local weighting scheme. The proposed model is then a well-calibrated GP/BNN hybrid. Finding BNNs that are well calibrated is still an open question and an important field of research.

Strengths: The paper is well executed and well written.

Weaknesses: The experiments are limited, and do not demonstrate improved performance.

Correctness: Everything looks to be correct

Clarity: Paper is well written and clear, the appendix is excellent

Relation to Prior Work: Yes

Reproducibility: Yes

Additional Feedback: Post-response update. The author response adressed my concerns well. The paper does have a nice contribution, but its experimental section is still unfortunately weak. I am raising my score to 5. ---- The paper works in the context of nested BNN’s, where latent node variables are mapped into probabilistic weights via NNs. The paper proposes to change the NN mapping into a GP mapping. The idea is motivated by desire to have more compact weight representation, and non-stationary weights. The benefit or motivation of either goal is not well defined. What problem is the GP or the input-dependent GP solving over the conventional BNN variant? The paper then describes two very different kinds of models arising from the use of GPs. The MetaGP couples hidden nodes and their weights together in a smooth way. It is difficult to see why this would be useful in the BNN framework, or what this means in terms of learning. The LocalMetaGP on the other hand is a clever model since it does provide the BNN with GP’s extrapolation properties: it reverts back to prior away from training data. This is likely to lead to safer uncertainties. On the other hand, neural network can learn to extrapolate due to being primal methods, and thus the GP’ness would remove this. Notably the GP’ness would make the model be far worse in transfer learning (eg. domain adaptation, multi-task learning, few-shot learning, extreme-learning, etc.) where GPs conventionally face lots of challenges. The fig4 is a good example of this, where the NN variants are able to learn the vertical trend of the orange cluster better than the GP variant. The main problem of the paper is that neither of the two new methods seems to improve upon earlier approaches. The MetaGP seems comparable to other BNN methods, and the LocalMetaGP seems to behave exactly like a GP would in all experiments. Since comparisons to vanilla GPs or deep GPs is lacking, its unclear if the LocalMetaGP has any benefits. Also note the duvenaud’s CKS line of methods for the kernel learning. The key question is then why not use (deep)GPs directly (which also have been scaled to billion-scale data)? It is also surprising that the only benchmark experiment is the OOD case, why not present standard MNIST or UCI benchmarks? What about large-scale prediction? The paper presents a clever and novel new GP-BNN hybrid architecture, with interesting and beneficial features. Unfortunately the new model is not demonstrated to surpass any close competitor, while the experiments are also too limited. Minor comments o the “f” as a random variable in the plate diagram is wrong: it should be placed between z and w (and perhaps more conveniently to label the edge) o the N(a|b) is poor notation due to the variance/mean being unclear o The eq 88 should make it more clear that f is a function instead of a finite function vector or evaluation. This is the main drawback of GPs, since it incurs quadratic/cubic scaling over the number of features. This can also be seen as a massively multi-task GP over w’s as the “tasks”. It seems odd not to mention this here, while the small-scaleness of the inputs is mentioned. o The local kernel of 110 is seemingly not used at all, but instead the inputs are transformed with a single NN layer. Please specify what is the final kernel used (either local or aux). o The example with periodic kernels is perhaps misplaced, since learning periodic kernels is notoriously difficult in general. If the data is periodic and one specifies the correct periodic kernel, the learning becomes overly easy. I feel the periodic examples confuse the paper’s message.

[Author Response · NeurIPS 2020]

We thank the reviewers for their time and their feedback. The reviewers highlight the interest of our method and find both the model and the experiments interesting and different from the literature. Criticism is focused on one key question: how is this model different from a GP, in particular in the case of a local model. In the following we address this question, clarify the structure of our model further, and answer individual points the reviewers raised.

(**R1, R2, R3, R4 - Representation & Advantages to GPs**) We propose a model which follows the flow: latent variables per unit->GP->weights and a local version of this which also includes a context variable as input to the GP. In the appendix, we explain that this model can be run per layer in two variants: either as a GP generating all weights in the network; per layer, with a per-layer kernel with individual kernel parameters and inducing points; or in a stacked fashion, where each layer's weights depend on the outputs of the layer below. In our global model, the parametrization of $k_w$ uses "inducing weights", the local model adds "inducing inputs" and makes the model distance-dependent. Depending on whether the model is used per layer, and whether the context variable is the input $x$ or the previous layer-activations, the model will have quite different applications and (dis)-advantages. When comparing to a GP, the closest variant may be when utilizing the local model conditioned on input x at all layers. Consider this thought-example to prove they differ even in that case: our model can -for an identical kernel choice as a GP- learn dramatically different functions than a GP since it can simply 'switch off' $k_{aux}$ (i.e. with a large lengthscale which sets all entries in $k_{aux}$ to 1) and revert to the global model $k_w$ which only utilizes the inducing weights but ignores the influence of the input. A GP can only express functions induced by the input-kernel, whereas our model can revert to a parametrization that ignores the influence of the input on the weights. This becomes more evident when considering per-layer or stacked parametrizations that blend with the $k_w$ kernel. We will add an example of this to the appendix. As such, the influence of $k_{aux}$ is an additional modeling tool to help induce biases into the prior and allow inheritance of beneficial properties of GPs when useful.

(**R1, R2, R3, R4 - Experiments**) We note that Deep Kernel Learning is a GP with a learned kernel that we compare to favorably. As per suggestions, we are adding comparisons to sparse GPs with RBF kernels to the supplement and will update Fig.16 (including notMNIST and fashionMNIST) to show the benefits of our model.

(**R1, R2, R3 - Scalability, Practicality**) When used per layer, the complexity is $O(|W| * M^2 + M^3)$ where $|W| =$ number of weights in a layer, M = number of inducing weights so it will be $O(|W|M^2)$ in practice. Empirically we found that we only require a small M, especially in a per-layer parametrization. The model thus induces an overhead compared to a regular mean field BNN, but maintains scalability in terms of size of the model.

(**R1, R3, R4 - Global Model & Modeling Benefits**) Another point of interest is what structure the latent-node parametrization of the global model induces and why this parametrization makes sense. Consider a per-layer kernel model, given samples z, the model is equivalent to a Matrix-Variate Normal weight model with components for weight columns and rows, parametrized by inducing weights. This has been shown to be useful in BNNs. Our global model adds two variations: (1) marginally over z, we have a mixture of matrix-variate Normal weight models as each 'unit' can change and (2) we share components between adjacent layers by sharing the unit-samples z. This can be seen as tying together the matrix variate distributions at each layer. Our construction can be used on any network structure, utilize convolution or sparse graph NNs with no notion of layers, we leave elaborate such constructions to future work.

(**R2**) Thanks for the helpful suggestions and comments. We will compare to ensembles in the next version.

(**R3**) **weight correlations:** conditionally on latent samples z, weights are independent when using a diagonal approximation. However, when z are marginalized out, correlations between all weights connected to a unit are still retained. We demonstrated this in Fig 11, App. G. **block diagonal:** Thanks for the suggestion. This is similar to the Partially Independent Training Conditional method in the GP literature. Our per-layer parametrizations are also moving in that direction and we intend to explore this suggestion further. **Budget vs model size:** our treatment of NNs is not focused so much on computational budget, as it is on capturing inductive biases and uncertainty which is about sample efficiency and robustness. **Fig.15** DKL is actually not the best method, Fig 15 shows our model to perform best.

(**R4**) **GP-ness:** We note that the GP predictions are different to that of the proposed method, e.g. for the dataset in Fig 4, the high-confidence regions for GPs are smaller. We also do not believe the GP-ness would hurt transfer learning in our formulation, but this goes beyond the scope and space we have in this paper. **Evaluation:** We provided some regression results on UCI in the appendix I.3, and we will add additional experiments on UCI and MNIST for some variants of our model and CIFAR-10 (last layer). We focus on the OOD setting since this is an advantage of our model and we compare against strong GP-baselines such as DKL. We'd love to see the community utilize some of the ideas regarding distance dependent weights demonstrated here in large scale applications down the line. **Minor:** $f$ is an entire function. We take the function space view, which is now widely adopted in the sparse GP literature, $w = f(z) + \epsilon$. **Line 110:** we used this kernel. For tasks with high dimensional inputs, an input warping can be used. **Periodic Kernel:** Thanks! It demonstrates that we can impose a controllable and non-trivial prior on weights, and periodicities in the past have been difficult to "cook into" NNs, see "Expressive priors in Bayesian neural networks" by Pearce et al. **CKL:** CKL is an exciting avenue to pursue in combination with our model to facilitate utilizing such rich priors for weight-based models.

[Meta-Review · NeurIPS 2020]

This work presents new and interesting ways to define a prior for Bayesian NNs by injecting a priori correlation structure in the weights. While there were reviewer questions about the motivation of such a technique (as opposed to using GPs in lieu of a neural network) the author response has convincingly addressed these concerns. There were also some concerns about interpretation of these functional priors, and how to control or tailor the inductive bias to a particular problem setting.